# Giant Left Atrial Appendage Aneurysm in a 6-Year-Old Girl with a Prothrombotic Genetic Predisposition: A Case Report and Literature Review

**DOI:** 10.3390/diagnostics15162070

**Published:** 2025-08-18

**Authors:** Horatiu Suciu, Emanuel-David Anitei, Valentin Ionut Stroe, Emilia Eleonora Brudan, Tudor Capilna, Hussam Al Hussein, Simina Ghiragosian, Paul Calburean, Mihaly Veres, Marius Mihai Harpa

**Affiliations:** 1Department of Surgery IV, George Emil Palade University of Medicine, Pharmacy, Science and Technology of Targu Mures, 540139 Targu Mures, Romania; horisuciu@gmail.com (H.S.); marius_harpa@yahoo.com (M.M.H.); 2Department of Cardiovascular Surgery, Emergency Institute for Cardiovascular Diseases and Transplantation Targu Mures, 540136 Targu Mures, Romania; vali.stroe@yahoo.com (V.I.S.); emilia.brudan@yahoo.com (E.E.B.); dodocapilna@yahoo.com (T.C.); 3Department of Anatomy and Embriology, George Emil Palade University of Medicine, Pharmacy, Science and Technology of Targu Mures, 540139 Targu Mures, Romania; alhussein.hussam@yahoo.com; 4Department of Pediatrics III, George Emil Palade University of Medicine, Pharmacy, Science and Technology of Targu Mures, 540139 Targu Mures, Romania; simina_r88@yahoo.com; 5Department of Biostatistics and Medical Informatics, George Emil Palade University of Medicine, Pharmacy, Science and Technology of Targu Mures, 540142 Targu Mures, Romania; calbureanpaul@gmail.com; 6Department of Anesthesia and Intensive Care, Emercency Institute for Cardiovascular Diseases and Transplantation of Targu Mures, 540136 Targu Mures, Romania; v_misy@yahoo.com

**Keywords:** left atrial appendage, aneurysm, pediatric cardiac surgery, congenital heart diseases

## Abstract

**Background:** The term ‘left atrial appendage aneurysm’ (LAAA) has been recognized since 1962, when it was first described. It is an exceedingly rare pathology that can affect both adults and children. Often asymptomatic, it may be discovered incidentally. The anomaly consists of an exaggerated dilation of the primitive portion of the left atrium, resulting from pectinate muscle dysplasia or as a consequence of an obstructive lesion between the left atrium and the mitral valve. Surgical intervention represents a reliable strategy for preventing catastrophic complications such as stroke, thromboembolism, and rupture. This is a very rare condition, which is why we aimed to present a case report along with a review of the literature. **Case presentation**: We report the case of a 6-year-old asymptomatic girl in whom a giant left atrial appendage aneurysm was incidentally detected during a routine transthoracic echocardiogram, associated with a small atrial septal defect and a prothrombotic genetic profile. The aneurysm was successfully excised, and the atrial septal defect was closed. The postoperative course was uneventful, and the patient was discharged home on the 8th postoperative day. **Conclusions**: Left atrial appendage aneurysm is rare in children and often asymptomatic, yet it may be life-threatening due to stroke or thromboembolism. Fetal echocardiography may be considered in selected high-risk pregnancies, and routine postnatal assessment is advised, with surgical intervention recommended particularly for patients with risk factors for thrombus formation in the left atrium or its appendage.

## 1. Introduction

Although the anatomy of the left atrium (LA) and left atrial appendage (LAA) is well known, their full physiological significance within the cardiovascular system is still being explored. The LAA, a finger-like extension of the LA, shows significant variability in size, shape, and anatomical relationships. In addition to its mechanical role as a decompression chamber during periods of elevated left atrial pressure, such as during left ventricular systole, it also plays a neurohumoral role, being richly innervated by sympathetic and parasympathetic fibers and contributing to the secretion of atrial and brain natriuretic peptides [1,2]. Left atrial appendage aneurysm (LAAA) is an extremely rare condition. Since its first description by Parmley et al. in 1962 [3], only 177 cases have been reported, with just 18 occurring in the pediatric population. Transthoracic echocardiography is the most commonly used diagnostic modality; in addition, LAAA can also be diagnosed prenatally through routine fetal echocardiography [4]. Left atrial appendage aneurysm may be classified as congenital or acquired. Congenital forms result from dysplasia and the thinning of the pectinate muscles and atrial bands, leading to localized outpouching. Acquired LAAA typically arises from elevated left atrial pressure (e.g., mitral valve disease) or atrial wall weakening (e.g., tuberculous or syphilitic myocarditis). According to its relationship with the pericardium, LAAA is divided into intrapericardial and extrapericardial types. Extrapericardial LAAA is associated with a pericardial defect that allows the herniation of the appendage. Intrapericardial forms include aneurysms of the left atrial wall, the appendage itself, or multisaccular aneurysms of the atrial wall [5]. Symptoms typically appear more frequently in the second and third decades of life as the aneurysm increases in size. Clinical presentation ranges from asymptomatic cases to severe complications such as stroke, peripheral embolism, supraventricular and ventricular arrhythmias, or aneurysm rupture. In addition, atypical symptoms such as hiccups, hoarseness, and persistent cough have also been reported [6]. Surgical treatment is recommended even in asymptomatic patients to prevent potential complications such as stroke, peripheral ischemia, arrhythmias, left ventricular outflow tract obstruction, or aneurysm rupture. This is particularly indicated in symptomatic patients; those with arrhythmias, intracavitary thrombus, signs of compression on adjacent structures, or increased thrombotic risk; or when concomitant cardiac surgery is planned. Various surgical techniques have been described, ranging from conventional sternotomy to minimally invasive endoscopic and off-pump approaches, as well as interventional procedures, all showing favorable outcomes [7,8]. We report the case of a 6-year-old girl with a giant left atrial appendage aneurysm, presenting with mild fatigue, diagnosed by echocardiography, and successfully treated with a surgical excision of the aneurysm, with a favorable postoperative course.

Due to the rarity of this condition and the limited data available in the literature, the standardization of treatment or development of clear surgical or medical management protocols remains challenging. We conducted a literature review on LAAA in children, with no time restrictions, to which we add the present case. This study aims to contribute new information and to consolidate the existing cases and treatment strategies, in order to support and facilitate the management of these patients.

## 2. Case Presentation

We present the case of a 6-year-old patient with normal overall development appropriate for her age, entirely asymptomatic, with no history of cyanosis, syncope, or exercise intolerance. A routine transthoracic echocardiogram performed in a local hospital incidentally revealed a cystic cardiac formation. Additionally, no other cases of cardiac malformations have been identified from a family genetic reservoir perspective. The electrocardiogram reveals a sinus rhythm with a heart rate of 90 bpm, with negative T waves in the precordial leads. The chest X-ray raises the suspicion of a tumor formation in the mediastinum and left hiliar region. Thoracic CT angiography revealed an aneurysm measuring approximately 6.5 × 4 cm at the level of the left atrial appendage (Figure 1).

Transthoracic echocardiography provides the definitive diagnosis, revealing a giant aneurysm of the left atrial appendage approximately 6 cm in size, with a 13 mm communication to the left atrium, a small atrial septal defect, normal cardiac contractility, and no valvular pathology (Figure 2).

Routine blood tests were normal. Genetic testing identified heterozygous MTHFR C677T, a heterozygous Factor XIII variant, and homozygous PAI-1 4G/4G, findings generally regarded as prothrombotic factors of low-to-moderate effect size. Due to the presence of these factors, which increase homocysteinemia, reduce clot stability, and decrease fibrinolysis, we considered that, in the setting of a large left atrial appendage aneurysm, they may act additively and increase the risk of intracavitary thrombosis and distal embolization, with additional potential complications including arrhythmias, compression, and rupture. Given the aneurysm size and the compounded thromboembolic risk, delaying surgery was deemed unlikely to be beneficial, and early operative management was pursued. In March 2025, a left atrial appendage aneurysm excision was performed under cardiopulmonary bypass, with cardiac arrest to close the atrial septal defect. Intraoperatively, a 6 × 4 cm aneurysm of the left atrial appendage was identified, with thin and friable walls. No thrombus was detected within the aneurysm (Figure 3). Additionally, a secundum-type atrial septal defect measuring approximately 1.5 cm was observed and closed.

No unexpected events occurred, and the patient remained hemodynamically and respiratorily stable. She was extubated within the first 6 h and had a favorable postoperative course, with a small pericardial effusion of no clinical significance. She was discharged home on postoperative day 8. Histopathology revealed areas of atrial muscle interspersed with areas of fibrous tissue. Follow-up echocardiography confirmed the absence of the dilated area in the left atrium, with no flow changes in the pulmonary veins or mitral valve.

## 3. Materials and Methods

This systematic review was conducted in accordance with the PRISMA 2020 guidelines (Figure 4). A comprehensive search was performed in PubMed, Web of Science, and the Cochrane Library with no date restrictions, using the keywords “left atrial appendage” AND “appendage aneurysm” OR “LAAA” AND “left atrial appendage aneurysm in pediatrics”. Only human studies involving pediatric patients (<18 years of age) were included. Case reports and case series were eligible for inclusion. In studies labeled as “case report and literature/systematic review,” only the original case was considered to avoid duplication. Review articles were excluded for the same reason. All references were imported into Covidence (Veritas Health Innovation, Melbourne, Australia) for screening and management. Two authors (E.-D.A. and P.C.) independently screened all titles, abstracts, and full texts. Disagreements were resolved by discussion and consensus. A total of 642 articles were identified. After removing duplicates and applying the inclusion and exclusion criteria (age >18 years, animal studies, inaccessible full text, or a lack of direct reference to LAAA), 64 articles were included in the final analysis. Data were extracted using a standardized form, including the following variables: first author, year of publication, sex, age, clinical presentation, prenatal diagnosis, associated conditions, arrhythmias, aneurysm dimensions and neck size, aneurysm type (congenital, acquired, intrapericardial, extrapericardial), imaging modalities (chest X-ray, TTE, TEE, CT, MRI, cardiac catheterization/angiography), presence of intracavitary thrombus, type of intervention (on-pump, off-pump, sternotomy, thoracotomy, minimally invasive), postoperative complications, and whether conservative treatment was chosen. All extracted data are summarized in Appendix A. Descriptive statistics were used for analysis. Categorical variables were presented as absolute numbers and percentages. Fisher’s exact test was used to evaluate the association between intracavitary thrombus and arrhythmias. Statistical analysis was performed using GraphPad Prism version 10.0 (GraphPad Software, Boston, MA, USA).

## 4. Results

We identified *n* = 642 articles in the PubMed, Web of Science, and Cochrane databases, with no time restrictions. After removing duplicates, each article was screened individually. Articles were excluded based on the following criteria: patient age over 18 years, lack of accessible or relevant information, and an absence of direct reference to the target pathology. Following this process, *n* = 64 articles were included for detailed analysis. In total, we describe 69 pediatric cases, as two of the included articles were case series, to which we add our own case. Among the *n* = 69 pediatric patients included in the analysis, the maximum age was 17 years and the minimum age was 1 day. The mean age was 5.8 ± 5.5 years (Figure 5).

Male sex was slightly predominant, with *n* = 34 cases (49.3%), while *n* = 31 patients (44.9%) were female. In *n* = 4 cases (5.8%), the sex was not specified. Symptom data were available for *n* = 66 of the *n* = 69 pediatric patients. In *n* = 3 cases, the clinical presentation was not specified. Among the *n* = 66 patients with reported symptoms, *n* = 30 (45.5%) were asymptomatic at the time of diagnosis. Varied respiratory symptoms, including dyspnea, respiratory distress, tachypnea, apnea, and bronchitis, were documented in 21 patients (31.8%). There were 16 patients (24.2%) with neurological symptoms, including stroke in 7 cases (10.6%), syncope in 4 (6.1%), convulsions in 1 (1.5%), dizziness in 3 (4.5%), and fainting in 1 (1.5%). Palpitations were noted in *n* = 8 (12.1%). Hiccups were described in a single case (1.5%). In one patient (1.5%), associated renal manifestations were reported, including hematuria and renal infarction (Figure 6).

Intracavitary thrombus was found in 10/69 patients (14.5%); 33.3% of those with arrhythmias (6/18) had thrombus, versus 7.8% of those without arrhythmias (4/51). This suggests a strong association between arrhythmic events and thrombus formation in LAAA (*p* = 0.0158)—see Table 1.

Prenatal identification was reported in *n* = 8 of the *n* = 69 cases, with gestational ages ranging from 19 to 31 weeks. No associated anomalies were reported in *n* = 33 cases (47.8%), and, in *n* = 2 patients (2.9%), this information was not available. The most frequent finding was mitral regurgitation, present in *n* = 13 patients (18.8%), followed by left ventricular compression in *n* = 6 cases (8.7%) and atrial septal defect in another *n* = 6 cases (8.7%). Patent ductus arteriosus and LAD compression were each described in *n* = 3 patients (4.3%). Two patients (2.9%) had anomalous coronary artery courses, one involving the circumflex artery and the other the left anterior descending artery. Thrombophilia was described in a single patient, *n* = 1 (1.5%). Additional associated findings included right ventricular involvement, left ventricular hypertrophy, pericardial effusion, pneumonia, Treacher Collins syndrome, mitral valvuloplasty, severe left ventricular dysfunction, foramen ovale permeability, and transposition of the great arteries. Atrial fibrillation (AF) was the most frequent arrhythmia, reported in *n* = 9 patients (13.0%), while atrial flutter was observed in *n* = 2 cases (2.9%). LAAA dimensions varied widely across the reported cases, and consistent quantification was challenging due to heterogeneous measurement methods. However, the smallest documented dimension was 23 × 10 mm, while the largest reached up to 200 mm in length. In one case, the aneurysm was considered acquired. In another, the etiology was not specified. Additionally, in two reports, the location of the aneurysm, intrapericardial or extrapericardial, was not mentioned, while two cases were explicitly described as extrapericardial. Transthoracic echocardiography was the most frequently used imaging modality, performed in *n* = 61 out of *n* = 69 cases (88.4%), followed by chest X-ray in *n* = 54 cases (78.3%), computed tomography (CT) in *n* = 26 cases (37.7%), transesophageal echocardiography (TEE) in *n* = 12 cases (17.4%), and cardiac catheterization or angiography in *n* = 10 cases (14.5%). In the reviewed cases, thrombus was identified in *n* = 10 patients (14.5%), while no thrombus was found in *n* = 48 cases (69.6%). In *n* = 5 cases (7.2%), the presence of spontaneous contrast was noted, and, in *n* = 6 cases (8.7%), the presence or absence of thrombus was not clearly specified. Surgical resection was the most common treatment approach, performed in 82.6% of cases, followed by stapled resection (2.9%), surgical plication (1.4%), and interventional closure with an Amplatzer device (1.4%), while no treatment was reported in 8.7% of cases and the management approach was unspecified in 2.9%. Cardiopulmonary bypass (CPB) was utilized in *n* = 32 cases (46.4%), sternotomy was performed in *n* = 39 cases (56.5%), thoracotomy in *n* = 7 cases (10.1%), and, in one case, a minimally invasive endoscopic approach was employed. Out of *n* = 69 cases, postoperative complications were reported in *n* = 9 patients (13.0%), including atrial flutter, pericardial effusion (*n* = 2 cases), residual aneurysm with left main compression, left ventricular pseudoaneurysm, right atrial thrombus, and one case requiring ECMO support due to left ventricular insufficiency following suture-related circumflex distortion. Conservative treatment was applied in *n* = 10 out of *n* = 69 cases (14.5%), including medical therapy with digoxin and amiodarone, the management of supraventricular tachycardia, and one case initially treated conservatively followed by surgery after 30 years.

## 5. Discussion

Left atrial appendage aneurysm (LAAA) was first described by Parmley in 1962. Approximately 180 cases have been reported in the literature to date, across all age groups, with a mean age of 30  ±  20 years (range: fetal age 28 weeks to 88 years). Most patients (24.8%, 25/101) were in their third decade of life, likely reflecting a progressive enlargement of the aneurysm over time. Although some studies have suggested no significant gender predominance, others reported a slightly higher incidence in female patients (53/101, 52.5%) compared to males (45/101, 44.6%) [3,9]. In our study, a slight male predominance was observed, with 49.3% male and 44.9% female patients. The age of the patients ranged from 1 day to 17 years, with a mean age of 5.8  ±  5.5 years. These findings are comparable to those reported by Norozi K. et al. [10] in their study on LAAA in children, which showed that approximately 48% of patients were male, 39% were female, and the mean age was 5.72  ±  5.69 years. Prenatal identification offers significant advantages in planning and guiding therapeutic management. The early detection of this pathology may prevent the development of associated complications. In our study, only eight cases were diagnosed prenatally, seven of whom underwent surgical treatment within the first months of life. Left atrial appendage aneurysms are extremely rare and can be identified as early as the prenatal period through fetal echocardiography. In a study by Xin Wang MD et al. [4], among 48,235 fetal echocardiograms, only 17 fetuses were found to have atrial aneurysms, of which 8 involved the left atrial appendage. Aneurysms may occur in either atrium, but postnatal data suggest that left atrial appendage aneurysms are more frequently encountered than right-sided ones, with a reported ratio of approximately 10:3 [11]. The left atrial appendage (LAA) has a distinct embryologic origin, developing from the primitive atrium around the fourth week of gestation. Its internal surface is trabeculated due to the presence of pectinate muscles. Congenital LAAA is attributed to dysplasia and the thinning of these pectinate muscles, leading to a localized outpouching of the LAA wall. Acquired LAAA may result from chronically elevated left atrial pressure—often secondary to mitral valve disease, atrial tachyarrhythmias, or inflammatory conditions such as tuberculous or syphilitic myocarditis. The left atrial appendage (LAA) has a distinct embryologic origin, developing from the primitive atrium around the fourth week of gestation. Its internal surface is trabeculated due to the presence of pectinate muscles. Congenital LAAA is attributed to dysplasia and the thinning of the pectinate muscles and associated atrial muscle bands, potentially linked to genetic syndromes such as RASopathies. Acquired LAAA may result from chronically elevated left atrial pressure—often secondary to mitral valve disease, atrial tachyarrhythmias, or inflammatory conditions such as tuberculous or syphilitic myocarditis [12,13]. Most left atrial appendage aneurysms are isolated cardiac anomalies. In only a minority of cases are they associated with other congenital defects requiring surgical correction. In the study by Aryal MR et al. [14], which included 82 patients, only 5 had associated congenital anomalies: 3 with atrial septal defect (ASD), 1 with ventricular septal defect (VSD), and 1 with Noonan syndrome. In our cohort, 6 patients had associated ASD, 3 had a persistent ductus arteriosus, 2 had coronary artery anomalies with abnormal courses, 1 case was associated with Treacher Collins syndrome, 1 with the transposition of the great arteries, and another with a complex congenital malformation. Due to the limited data and scarce research, this pathology cannot currently be linked to any specific genetic syndrome [15]. Patients are often asymptomatic, and symptoms typically emerge in the second or third decade of life, likely due to the progressive enlargement of the aneurysm over time. As a result, most diagnoses occur during this period. The clinical presentation may also vary depending on associated anomalies. In fact, congenital LAAA can occur in isolation or be associated with other congenital defects, as previously mentioned. In our study, approximately half of the patients were asymptomatic. Among symptomatic cases, respiratory symptoms were the most frequently reported, followed by neurological manifestations and arrhythmias. Less specific symptoms such as cough, hoarseness, or hiccups may arise due to the compression of adjacent structures or mechanical irritation of the phrenic nerve [5,9,16]. Neurological symptoms can be catastrophic; in our cohort, stroke was the initial presentation in seven patients, with a mean age of 10.43 ± 5.47 years. Of these, only two had atrial fibrillation, and four had evidence of intracavitary thrombus. These findings underscore the potential benefit of routine echocardiographic screening, including fetal echocardiography, to allow early diagnosis and management and prevent such complications. Peripheral embolism has also been described. One patient in our study presented with hematuria and renal infarction. Palpitations and chest pain may also occur. A potential mechanism is coronary artery compression. Three patients in our series had a compression of the left anterior descending artery, and all three presented with arrhythmias, including atrial tachycardia, atrial fibrillation, and atrial flutter [10,17]. Supraventricular and ventricular arrhythmias may result from conduction tissue stretch or from the compression of the left coronary artery in cases of large aneurysms. In some patients, the arrhythmic focus originates directly from the aneurysmal left atrial appendage itself. This has been demonstrated in electrophysiological studies, where the aneurysmal LAA served as the substrate for inducible, sustained intra-atrial reentry tachycardia. In one reported case, ablation was not pursued, as a surgical resection of the aneurysm was considered the optimal strategy to eliminate both the arrhythmogenic focus and the potential source of thromboembolic events [18,19]. Although routine laboratory parameters were within normal limits, targeted genetic analysis revealed three thrombophilia-associated variants: heterozygous MTHFR C677T, heterozygous Factor XIII, and homozygous PAI-1 4G/4G. While each variant individually is generally considered to confer a low-to-moderate prothrombotic risk, the coexistence of these factors with a giant left atrial appendage aneurysm (LAAA) may exert a synergistic effect in amplifying the prothrombotic risk. The MTHFR C677T variant is associated with hyperhomocysteinemia, a recognized contributor to endothelial dysfunction and thrombogenesis. The Factor XIII variant may impair fibrin cross-linking and clot stabilization, whereas the homozygous PAI-1 4G/4G genotype leads to reduced fibrinolytic activity due to elevated plasminogen activator inhibitor levels. In combination, these abnormalities may substantially increase the likelihood of intracavitary thrombus formation and subsequent systemic embolization. Despite the absence of clinical symptoms, arrhythmias, or imaging evidence of thrombus, the substantial aneurysmal size in conjunction with this procoagulant genetic profile provided a compelling indication for surgical intervention. This approach aimed to prevent potentially life-threatening complications, including thromboembolic events, aneurysmal rupture, coronary artery compression, and malignant arrhythmias. Although conservative management with anticoagulation has been described in selected cases, the scarcity of long-term follow-up data, the absence of standardized treatment protocols, and the uncertainty regarding the natural history of LAAA in the context of hereditary thrombophilia, particularly in pediatric patients, limit the ability to recommend this approach with confidence. In this context, we considered that surgical resection represented a reasonable and potentially definitive option, aiming to reduce the risk of severe complications while acknowledging the need for further evidence to establish optimal management strategies. Dimensions vary considerably. The largest aneurysm reported in an adult measured 18 cm × 10 cm × 8 cm. Larger aneurysms are more frequently associated with intracavitary thrombi, especially in patients with arrhythmias such as atrial fibrillation or flutter, or in those with a genetic predisposition to prothrombotic states. In our study, the reported dimensions varied due to differences in imaging modalities, including transthoracic echocardiography, CT angiography, and cardiac MRI. When available, we considered intraoperative measurements or the largest values recorded. The largest aneurysm identified measured 200 mm and contained an intracavitary thrombus, while the smallest measured 23 × 10 mm [20]. Left atrial appendage aneurysms (LAAAs) can be classified based on their etiology and their anatomical relationship to the pericardium. From an etiological perspective, LAAAs are either congenital, caused by dysplasia and the thinning of the pectinate muscles and associated atrial muscle bands leading to localized outpouching of the left atrial wall or appendage, or acquired, typically resulting from chronically elevated left atrial pressure as seen in mitral valve disease or following inflammatory conditions such as tuberculous or syphilitic myocarditis [21]. Anatomically, LAAAs are described as intrapericardial or extrapericardial. Extrapericardial aneurysms are commonly associated with a pericardial defect, allowing the left atrial appendage to herniate and progressively enlarge over time. Despite advances in imaging techniques, the distinction between intrapericardial and extrapericardial forms is most accurately made intraoperatively. Intrapericardial aneurysms may arise from the left atrial wall or the left atrial appendage, or present as multisaccular formations of the atrial wall [22,23]. One patient in our study was classified as having acquired LAAA, while two patients were identified with extrapericardial LAAA. An important aspect to consider in patients with LAAA is the presence of concomitant mitral regurgitation. In our study, 13 patients (18.8%) had associated mitral regurgitation, moderate in 5 cases and severe in 4. Considering the left atrium and the left atrial appendage as two anatomically distinct but closely related structures, the aneurysms arising from each can be differentiated. LAAAs are more frequently reported and more commonly contain intracavitary thrombi. In contrast, left atrial aneurysms (LAAs) are rarer, typically located on the posteroinferior wall of the left atrium, and, due to their proximity to the mitral annulus, they may more often lead to mitral regurgitation. Nonetheless, the clinical presentation and potential complications of both entities tend to overlap [24,25]. Two of these patients also had a mitral valve cleft, which may support an acquired etiology for the aneurysm. In most of the remaining cases, the regurgitation was attributed to the compression of the left ventricle caused by the aneurysm. Notably, surgical resection resulted in an improvement to mitral valve function. There are several types of atrial appendage aneurysms, including left atrial appendage aneurysm, right atrial appendage aneurysm (RAAA), and bilateral atrial appendage aneurysms, with LAAA being the most frequently reported. The criteria for congenital intrapericardial LAAA include origin from a morphologically normal left atrium, clear communication with the atrial cavity, intrapericardial location, and distortion of the left ventricular free wall [7,26]. In our case, the patient had an intact pericardium and a well-demarcated left atrium clearly separated from the aneurysmal structure of the left atrial appendage. No significant associated pathologies were identified that could have contributed to the development of the aneurysm. Therefore, the aneurysm was classified as a congenital intrapericardial LAAA. In young infants, marked cardiomegaly on chest radiography may raise suspicion for various underlying conditions, such as congenital heart defects, cardiomyopathies, perinatal myocarditis, metabolic disorders (including glycogen and lysosomal storage diseases), mitochondrial disorders, cardiac tumors, and, more rarely, a left atrial appendage aneurysm [27,28]. Transesophageal echocardiography (TEE) has higher sensitivity than transthoracic echocardiography (TTE) in detecting LAAA (90% vs. 45%) and is especially valuable intraoperatively for identifying thrombi and evaluating surgical results, particularly in minimally invasive procedures [6,29]. TTE, though less sensitive, is widely available, non-invasive, and effective in diagnosing LAAA, especially using the subxiphoid view to visualize aneurysm size, its communication with the atrium, and surrounding structures. In pediatrics, an LAA enlarged 1.5 times above normal may indicate aneurysm [14,30]. Cardiac CT and MRI are valuable tools in evaluating LAAA, offering detailed information on coronary anatomy and the aneurysm’s relationship to adjacent structures. They are also essential in the differential diagnosis, which includes pericardial cyst, LV pseudoaneurysm, dilated coronary sinus, cardiac tumor, mediastinal mass, and pericardial or extracardiac fluid collection. In pediatric patients, imaging often requires sedation or anesthesia, which may carry additional risks [31,32]. Surgical resection is recommended even in asymptomatic patients to prevent serious complications such as stroke, peripheral embolism, arrhythmias, left ventricular outflow tract obstruction, or aneurysmal rupture. One of the earliest cases reported in the literature describes a 9-year-old boy who developed symptoms, including syncope, exertional dyspnea, and palpitations. Due to limited imaging capabilities at that time, an exploratory thoracotomy was performed, revealing a left atrial appendage aneurysm that was not resected intraoperatively because of rhythm disturbances and hemodynamic instability. While awaiting re-intervention, the patient developed refractory atrial fibrillation, peripheral embolism, and stroke. Surgical resection was eventually performed with complete neurological recovery. This case underscored the importance of early surgical intervention to prevent potentially devastating complications. Surgery is particularly indicated in symptomatic patients; those with arrhythmias, intracavitary thrombus, signs of compression on adjacent structures, or thrombophilic conditions; or when concomitant cardiac procedures are planned [3,33,34]. Although not generally recommended, medical therapy may be considered in selected cases where patients decline surgical intervention. Conservative management, including antiarrhythmic therapy, may be an option for patients without intracavitary thrombus and with small, stable aneurysms. Plonska-Gosciniak et al. [35] reported a case of LAAA associated with supraventricular arrhythmia that remained stable under medical treatment over a 20-year follow-up. Similarly, Valentino et al. [36] described a patient who initially opted for conservative management but later required surgical resection after significant aneurysmal enlargement over five years. In our cohort, one patient declined surgery and received antiarrhythmic therapy, with follow-up imaging showing a reduction in aneurysm size. This case suggests that antiarrhythmic drug therapy may serve as a temporary strategy in selected pediatric patients without intracavitary thrombus, potentially delaying the need for surgical intervention. The surgical management of LAAA should be adapted to the patient’s anatomy and clinical risk profile. In cases without intracavitary thrombi or associated malformations requiring correction, resection may be performed safely without cardiopulmonary bypass through a left lateral thoracotomy, guided by transesophageal echocardiography. This approach helps to preserve left atrial volume and function. When the aneurysm is large or contains thrombi, or when the anatomy is complex, the use of cardiopulmonary bypass is recommended to reduce the risk of embolic events and ensure complete and safe excision. The intraoperative visualization of the coronary arteries, particularly the circumflex artery and any anomalous coronary courses, as well as pulmonary veins and their drainage, is essential. Transesophageal echocardiography provides real-time guidance during resection and supports precise surgical decision-making [37]. An interesting case has been described in which a distortion of the circumflex artery, most likely caused by surgical suturing, led to left ventricular dysfunction, requiring mechanical circulatory support. Tension and traction on the atrial tissue should be avoided, as they can compromise atrial geometry. If minimal atrial tissue remains after resection and direct suturing leads to deformation, atrial reconstruction with a heterologous pericardial patch may be necessary [38]. An interventional approach has also been reported using an Amplatzer occluder device placed at the aneurysmal orifice. Kothandam S. et al. [39] described a 17-year-old patient with a history of stroke in whom the aneurysm was successfully excluded using this minimally invasive technique. The classic method of aneurysm excision via sternotomy and cardiopulmonary bypass remains a commonly used and reliable technique in these patients. However, minimally invasive and endoscopic surgical approaches are increasingly being adopted. Several cases of LAAA have been successfully treated using these techniques, offering comparable outcomes with additional advantages such as excellent visualization, faster recovery, and superior cosmetic results [8,40]. The exclusion of the aneurysm using snare, stapling, or AtriClip techniques may be challenging and even hazardous in cases of large aneurysms with extremely fragile walls, despite their successful use in carefully selected cases. As noted by Burke and associates [41] regarding aneurysm stapling, such techniques should be approached with caution due to their potential risks. Sternotomy remains the most commonly employed surgical approach for LAAA repair, with more than 50% of procedures in our study performed using this technique. It offers optimal exposure, facilitates the safe identification of coronary artery anatomy, and ensures precise control during aneurysm exclusion or excision, which is particularly important when managing thin and friable aneurysmal walls. More than 40% of patients in our cohort underwent surgery with the use of cardiopulmonary bypass, including all cases with documented intracavitary thrombus. The use of bypass in these situations is essential to reduce the risk of embolization and to ensure procedural safety. Although minimally invasive and endoscopic techniques are increasingly utilized with good outcomes, they are best applied in carefully selected cases and by experienced teams, given the challenges associated with fragile aneurysmal tissue [42].

## 6. Conclusions

Left atrial appendage aneurysm is a rare pathology in the pediatric population, but it can be fatal due to its associated complications, such as stroke or thromboembolism. Given that most patients exhibit symptoms primarily in the second or third decade of life, and children are often asymptomatic, it is crucial to monitor pregnancies using fetal echocardiography followed by routine echocardiographic assessment. Surgical intervention is recommended for patients diagnosed with LAAA, particularly in those with predisposing factors for thrombus formation within the left atrium or left atrial appendage, such as atrial fibrillation, prothrombotic profile, or genetic mutations associated with increased thrombosis risk.

## 7. Limitations

This review has several limitations. First, no formal quality assessment of the included case reports and case series was performed, which may affect the consistency and reliability of the extracted data. Second, the small number of reported pediatric cases, along with the variability in clinical presentation, diagnostic methods, aneurysm size, and therapeutic approaches, limits the ability to compare cases and draw generalizable conclusions. These differences also made a meta-analysis unfeasible. Third, publication and selection bias must be considered, as only published cases were included. Finally, some reports lacked complete data on important clinical variables such as surgical technique, complications, or follow-up outcomes.

## Figures and Tables

**Figure 1 diagnostics-15-02070-f001:**
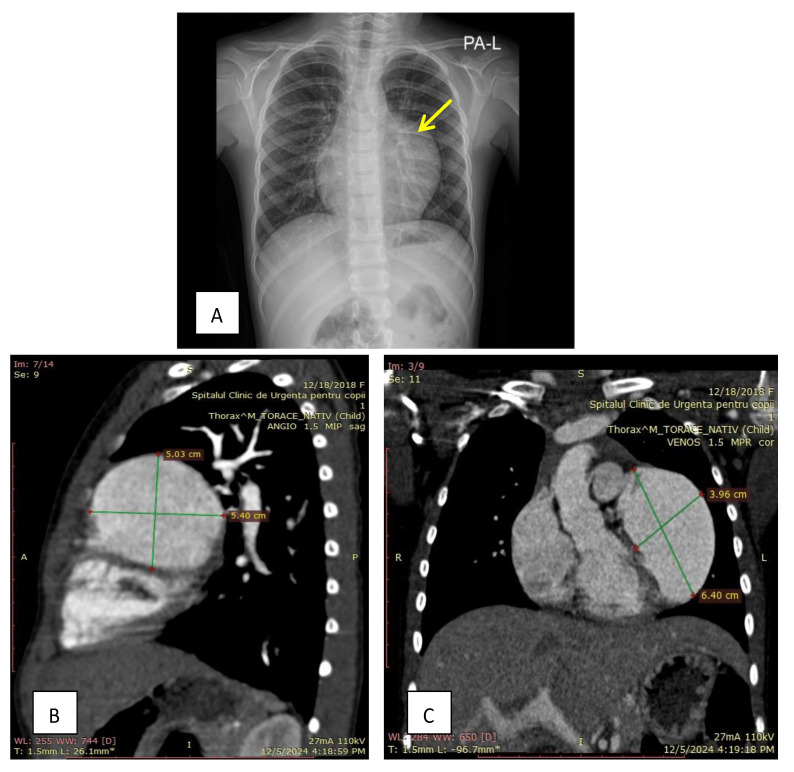
(**A**)—The chest X-ray revealed significant cardiomegaly. (**B**)—Sagittal thoracic CT angiography revealed a large aneurysm of the left atrial appendage. (**C**)—Frontal thoracic CT angiography revealed a large aneurysm of the left atrial appendage, measuring 6.40 cm by 3.96 cm.

**Figure 2 diagnostics-15-02070-f002:**
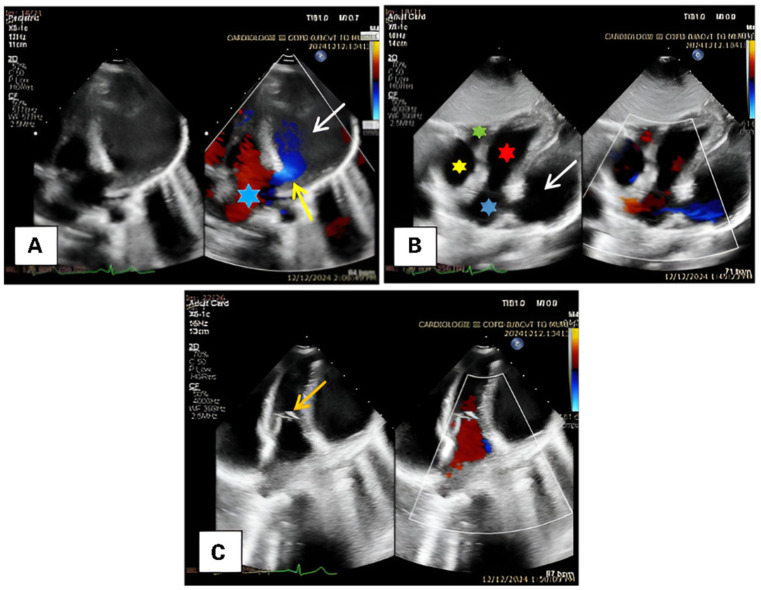
(**A**)—Transthoracic echocardiography showing the communication between the left atrium and the aneurysm (yellow arrow). The white arrow indicates the LAAA, and the blue star marks the LA. (**B**)—Five-chamber view. The yellow star indicates the right atrium, the green star the right ventricle, the blue star the left atrium, the red star the left ventricle, and the white arrow indicates the aneurysm. (**C**)—In the foreground is the mitral valve, which shows no regurgitation (orange arrow).

**Figure 3 diagnostics-15-02070-f003:**
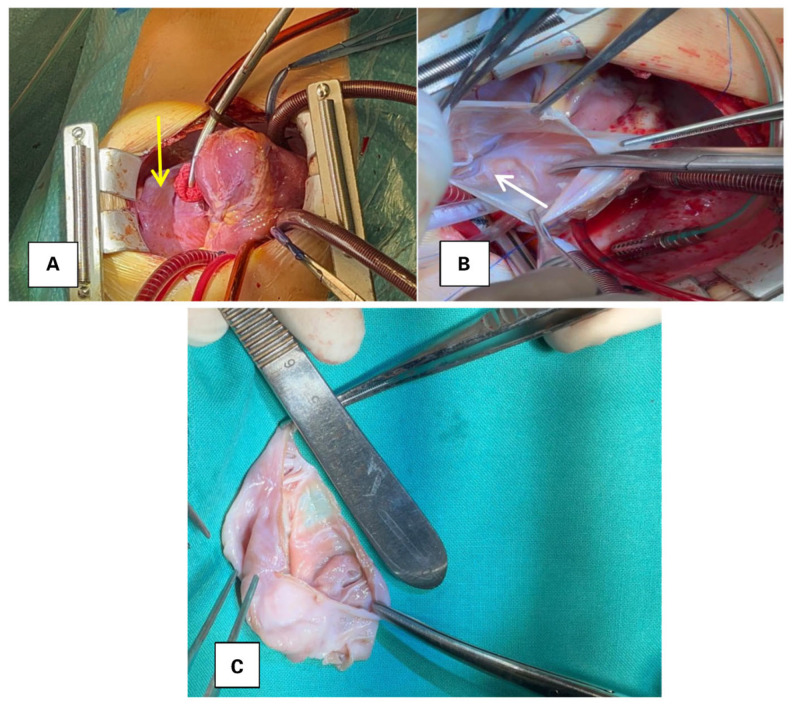
(**A**)—Intraoperative image showing the left atrial appendage aneurysm indicated by the yellow arrow. (**B**)—Opened LAAA, with the circumflex artery visible and indicated by the white arrow. (**C**)—Excised LAAA with thin and fragile walls.

**Figure 4 diagnostics-15-02070-f004:**
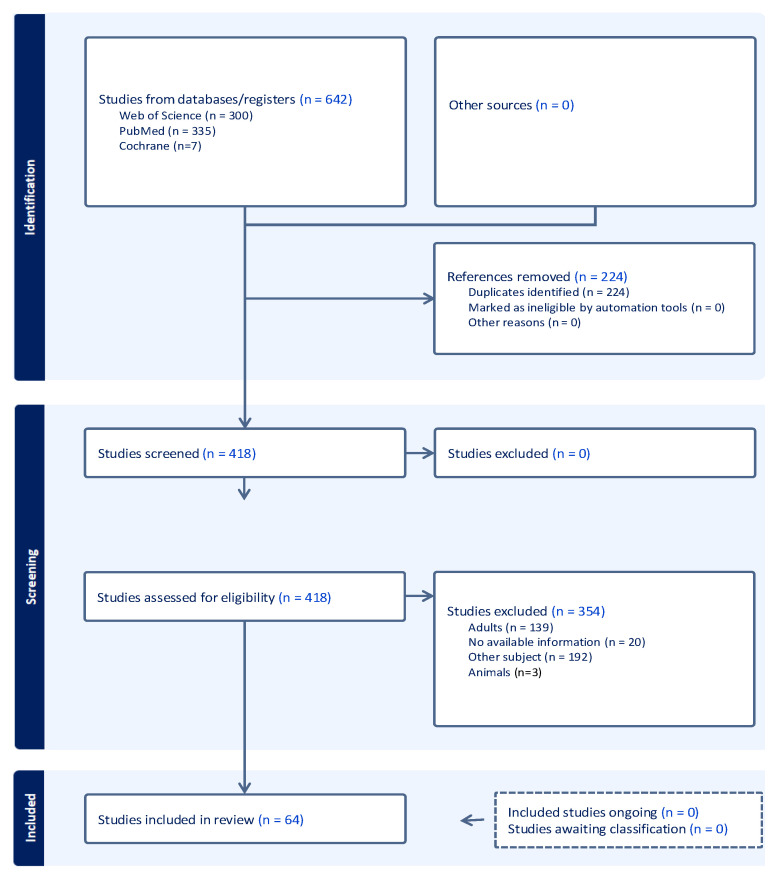
Prisma flow diagram.

**Figure 5 diagnostics-15-02070-f005:**
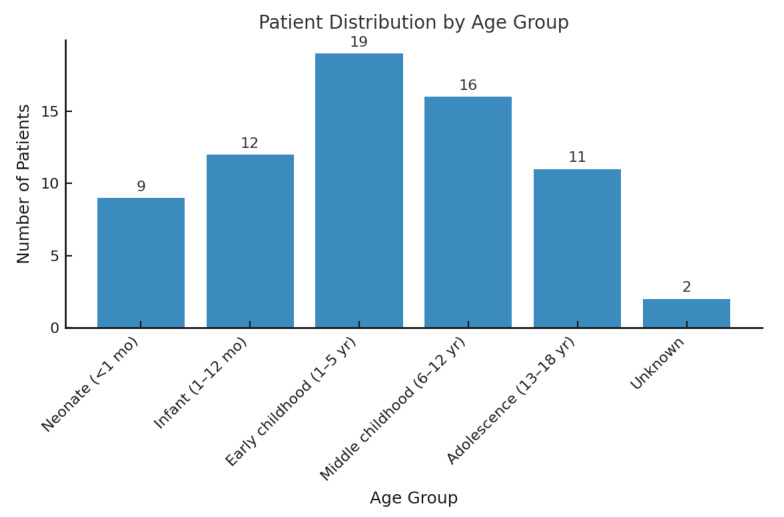
Patient distribution by age group.

**Figure 6 diagnostics-15-02070-f006:**
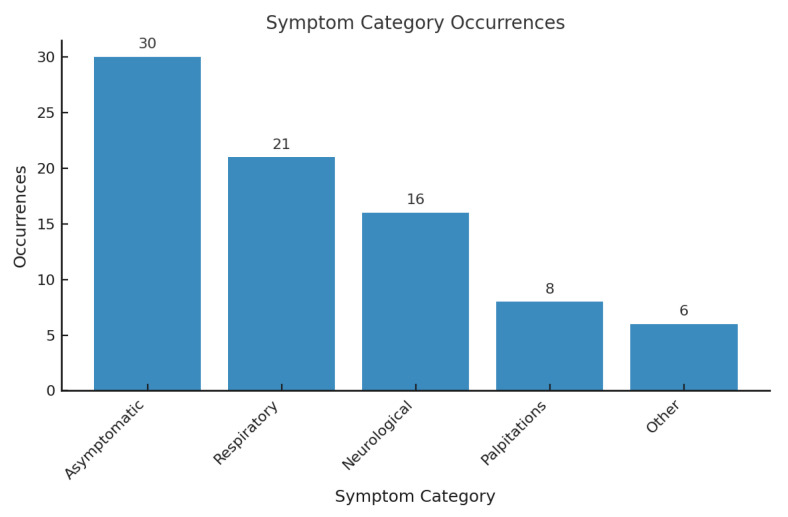
Symptom category occurrences.

**Table 1 diagnostics-15-02070-t001:** Associations between the presence of thrombus in left atrial appendage aneurysms and arrythmia.

Arrhythmia Status	No Thrombus, *n* (%)	Thrombus, *n* (%)	Fisher’s Exact *p* Value
Absent	47 (92.2%)	4 (7.8%)	
Present	12 (66.7%)	6 (33.3%)	0.0158

## Data Availability

Data availability statements are stored in the database of the Emergency Institute for Cardiovascular Diseases and Transplantation Targu Mures and are available upon request from the corresponding author.

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
