# Peer review of "Giant Left Atrial Appendage Aneurysm in a 6-Year-Old Girl with a Prothrombotic Genetic Predisposition: A Case Report and Literature Review"

_diagnostics, 2025, doi:10.3390/diagnostics15162070_

Round 1

Reviewer 1 Report

Comments and Suggestions for Authors

The article presents significant scientific interest, focusing on a rare and clinically important pathology – left atrial appendage aneurysm in the pediatric population. The combination of a detailed clinical case with a systematic literature review (n=69 pediatric cases) is a strength of the work. The study is relevant, well-structured, and contributes to the accumulation of data on this rare condition. However, some revisions are required to enhance the quality and clarity of the manuscript.

In the Abstract: The recommendation "essential to monitor pregnancies through fetal echocardiography followed by routine postnatal assessment" sounds very strong. Given the rarity of the pathology (only 8 prenatal diagnoses reported in the entire review), it would be advisable to soften the wording in the abstract (e.g., "could be considered" or "may be beneficial"), especially since the article lacks a cost/effectiveness analysis of this approach. A more balanced phrasing is present in the Conclusions section.

In the Case Description: It is recommended to briefly explain the clinical significance of the identified heterozygous MTHFR C677T and Factor XIII mutations, as well as the homozygous PAI-1 mutation, in the context of thrombophilia and their contribution to the decision-making process regarding surgery.

In the Materials and Methods section:

It is recommended to add information on who conducted the title/abstract screening and the full-text assessment for eligibility, and how it was done. Was there independent review by two authors with conflicts resolved by a third?

Describe which specific data were extracted from the included articles (e.g., demographics, symptoms, aneurysm size, associated anomalies, diagnostic methods, treatment, outcomes) and using what standardized form.

Considering that the review is primarily based on case reports and case series, it is worth briefly mentioning whether a formal assessment of their quality was performed and how this influenced the analysis. If not performed, this should be stated as a limitation.

Specify which statistical software package was used for the analysis.

Figure 6: It is recommended to align Figure 6 with the text of the article.

Author Response

1. Summary

2. Questions for General Evaluation

Reviewer’s Evaluation

Response and Revisions

Does the introduction provide sufficient background and include all relevant references?

Yes/Can be improved/Must be improved/Not applicable

-

Are all the cited references relevant to the research?

Yes/Can be improved/Must be improved/Not applicable

-

Is the research design appropriate?

Yes/Can be improved/Must be improved/Not applicable

-

Are the methods adequately described?

Yes/Can be improved/Must be improved/Not applicable

-

Are the results clearly presented?

Yes/Can be improved/Must be improved/Not applicable

-

Are the conclusions supported by the results?

Yes/Can be improved/Must be improved/Not applicable

-

3. Point-by-point response to Comments and Suggestions for Authors

Comments 1: In the Abstract: The recommendation "essential to monitor pregnancies through fetal echocardiography followed by routine postnatal assessment" sounds very strong. Given the rarity of the pathology (only 8 prenatal diagnoses reported in the entire review), it would be advisable to soften the wording in the abstract (e.g., "could be considered" or "may be beneficial"), especially since the article lacks a cost/effectiveness analysis of this approach. A more balanced phrasing is present in the Conclusions section.

Response 1: Thank you for this hopeful observation. You are right that our previous wording could imply a universal recommendation not supported by cost-effectiveness data. We have softened the language in the Abstract. “Left atrial appendage aneurysm is rare in children and often asymptomatic, yet it may be life-threatening due to stroke or thromboembolism. Fetal echocardiography may be considered in selected high-risk pregnancies, and routine postnatal assessment is advised, with surgical intervention recommended particularly for patients with risk factors for thrombus formation in the left atrium or its appendage.” (lines 36-40)

Comments 2: In the Case Description: It is recommended to briefly explain the clinical significance of the identified heterozygous MTHFR C677T and Factor XIII mutations, as well as the homozygous PAI-1 mutation, in the context of thrombophilia and their contribution to the decision-making process regarding surgery.

Response 2: We thank you for this valuable suggestion. We have expanded the informations from Case presentation. “Routine blood tests were normal. Genetic testing identified heterozygous MTHFR C677T, a heterozygous Factor XIII variant, and homozygous PAI-1 4G/4G, findings generally regarded as prothrombotic factors of low-to-moderate effect size. Due to the presence of these factors which increase homocysteinemia, reduce clot stability, and decrease fibrinolysis, we considered that, in the setting of a large left atrial appendage aneurysm, they may act additively and increase the risk of intracavitary thrombosis and distal embolization, with additional potential complications including arrhythmias, compression, and rupture. Given the aneurysm size and the compounded thromboembolic risk, delaying surgery was deemed unlikely to be beneficial, and early operative management was pursued.” (line 139-148)

Comments 3: It is recommended to add information on who conducted the title/abstract screening and the full-text assessment for eligibility, and how it was done. Was there independent review by two authors with conflicts resolved by a third?

Response 3: We appreciate your recomandation. We hope that we clarified this informations in section Materials and Methods. “This systematic review was conducted in accordance with the PRISMA 2020 guidelines (Figure 4). A comprehensive search was performed in PubMed, Web of Science, and the Cochrane Library with no date restrictions, using the keywords: “left atrial appendage” AND “appendage aneurysm” OR “LAAA” AND “left atrial appendage aneurysm in pediatrics”. Only human studies involving pediatric patients (<18 years of age) were included. Case reports and case series were eligible for inclusion. In studies labeled as “case report and literature/systematic review,” only the original case was considered to avoid duplication. Review articles were excluded for the same reason. All references were imported into Covidence (Veritas Health Innovation, Melbourne, Australia) for screening and management. Two authors (E.-D.A. and P.C.) independently screened all titles, abstracts, and full texts. Disagreements were resolved by discussion and consensus. A total of 642 articles were identified. After removing duplicates and applying the inclusion and exclusion criteria (age >18 years, animal studies, inaccessible full text, or lack of direct reference to LAAA), 64 articles were included in the final analysis. Data were extracted using a standardized form, including the following variables: first author, year of publication, sex, age, clinical presentation, prenatal diagnosis, associated conditions, arrhythmias, aneurysm dimensions and neck size, aneurysm type (congenital, acquired, intrapericardial, extrapericardial), imaging modalities (chest X-ray, TTE, TEE, CT, MRI, cardiac catheterization/angiography), presence of intracavitary thrombus, type of intervention (on-pump, off-pump, sternotomy, thoracotomy, minimally invasive), postoperative complications, and whether conservative treatment was chosen. All extracted data are summarized in Table 1 (Supplementary Material). Descriptive statistics were used for analysis. Categorical variables were presented as absolute numbers and percentages. Fisher’s exact test was used to evaluate the association between intracavitary thrombus and arrhythmias. Statistical analysis was performed using GraphPad Prism version 10.0 (GraphPad Software, Boston, USA).” (lines 179-205).

Comments 4: Describe which specific data were extracted from the included articles (e.g., demographics, symptoms, aneurysm size, associated anomalies, diagnostic methods, treatment, outcomes) and using what standardized form.

Response 4: Thank you, we described that in section Material and Methods (lines 179-205).

Comments 5: Considering that the review is primarily based on case reports and case series, it is worth briefly mentioning whether a formal assessment of their quality was performed and how this influenced the analysis. If not performed, this should be stated as a limitation.

Response 5: Thank you for highlighting this important aspect. We added a Limitations section. “This review has several limitations. First, no formal quality assessment of the included case reports and case series was performed, which may affect the consistency and reliability of the extracted data. Second, the small number of reported pediatric cases, along with the variability in clinical presentation, diagnostic methods, aneurysm size, and therapeutic approaches, limits the ability to compare cases and draw generalizable conclusions. These differences also made a meta-analysis unfeasible. Third, publication and selection bias must be considered, as only published cases were included. Finally, some reports lacked complete data on important clinical variables such as surgical technique, complications, or follow-up outcomes.” (lines 575-584).

Comments 6: Specify which statistical software package was used for the analysis.

Response 6: Thank you, we described that in section Material and Methods (lines 179-205).

Comments 7:  Figure 6: It is recommended to align Figure 6 with the text of the article.

Response 7: Thank you for the insightful observation. We have reconsidered the symptom classification and provided a more detailed description. “Varied respiratory symptoms including dyspnea, respiratory distress, tachypnea, apnea and bronchitis were documented in 21 patients (31.8%). There were 16 patients (24.2%) with neurological symptoms, including stroke in 7 cases (10.6%), syncope in 4 (6.1%), convulsions in 1 (1.5%), dizziness in 3 (4.5%), and fainting in 1 (1.5%).” (lines 241-245)

4. Response to Comments on the Quality of English Language

Response 1: Thank you for your comment

5. Additional clarifications

No additional clarifications

Reviewer 2 Report

Comments and Suggestions for Authors

The article presents a description of a rare case of giant left atrial appendage aneurysm in a 6-year-old girl with a prothrombotic genetic predisposition. A detailed clinical presentation is illustrated with informative CT and ultrasound images of the patient's heart. The clinical case is supplemented with a literature review. The structure of the manuscript contains sections, which are typical for systematic review, in particular the "materials and methods" and "results". This significantly complicates the understanding of the article. Thereby, I recommend the authors to present a narrative review in the Discussion section. The content of the preasented  literature review is seems to be similar to the article, recently published by Norozi K et al ( 10.3389/fcvm.2023.1211619). Since the title of the current article emphasizes prothrombotic genetic predisposition, I recommend to discuss the features of treatment for giant aneurysm in patients with hereditary thrombophilia.

Author Response

1. Summary

2. Questions for General Evaluation

Reviewer’s Evaluation

Response and Revisions

Does the introduction provide sufficient background and include all relevant references?

Yes/Can be improved/Must be improved/Not applicable

-

Are all the cited references relevant to the research?

Yes/Can be improved/Must be improved/Not applicable

-

Is the research design appropriate?

Yes/Can be improved/Must be improved/Not applicable

-

Are the methods adequately described?

Yes/Can be improved/Must be improved/Not applicable

-

Are the results clearly presented?

Yes/Can be improved/Must be improved/Not applicable

-

Are the conclusions supported by the results?

Yes/Can be improved/Must be improved/Not applicable

-

3. Point-by-point response to Comments and Suggestions for Authors

Comments 1: The article presents a description of a rare case of giant left atrial appendage aneurysm in a 6-year-old girl with a prothrombotic genetic predisposition. A detailed clinical presentation is illustrated with informative CT and ultrasound images of the patient's heart. The clinical case is supplemented with a literature review. The structure of the manuscript contains sections, which are typical for systematic review, in particular the "materials and methods" and "results". This significantly complicates the understanding of the article. Thereby, I recommend the authors to present a narrative review in the Discussion section. The content of the preasented  literature review is seems to be similar to the article, recently published by Norozi K et al ( 10.3389/fcvm.2023.1211619). Since the title of the current article emphasizes prothrombotic genetic predisposition, I recommend to discuss the features of treatment for giant aneurysm in patients with hereditary thrombophilia.

Response 1: We sincerely thank you for this constructive and valuable comment. We believe the current structure of the manuscript, which includes distinct sections for Materials and Methods and Results, offers clarity and methodological transparency, particularly considering the rarity of the condition and the need to synthesize data from multiple individual case reports. However, in order to facilitate a better understanding of the content, we have implemented several improvements throughout the manuscript, which are now reflected in the revised version. We also appreciate the reference to the excellent and timely article by Norozi et al. which served as a valuable point of reference and inspiration. The case described in that publication has also been included in our analysis. Nonetheless, our intention was to provide an updated and comprehensive synthesis of the pediatric literature on left atrial appendage aneurysm. Over the past five years, our review identified and included 22 pediatric cases, compared to the 16 reported in the aforementioned study. Furthermore, to our knowledge, the case we present is the only reported instance of a pediatric left atrial appendage aneurysm associated with a confirmed prothrombotic genetic predisposition. Also, we have incorporated additional information in the Discussion section regarding the management of patients with aneurysm and concomitant thrombophilia. “Although routine laboratory parameters were within normal limits, targeted genetic analysis revealed three thrombophilia-associated variants: heterozygous MTHFR C677T, heterozygous Factor XIII, and homozygous PAI-1 4G/4G. While each variant individually is generally considered to confer a low-to-moderate prothrombotic risk, the coexistence of these factors with a giant left atrial appendage aneurysm (LAAA) may exert a synergistic effect in amplifying the prothrombotic risk. The MTHFR C677T variant is associated with hyperhomocysteinemia, a recognized contributor to endothelial dysfunction and thrombogenesis. The Factor XIII variant may impair fibrin cross-linking and clot stabilization, whereas the homozygous PAI-1 4G/4G genotype leads to reduced fibrinolytic activity due to elevated plasminogen activator inhibitor levels. In combination, these abnormalities may substantially increase the likelihood of intracavitary thrombus formation and subsequent systemic embolization. Despite the absence of clinical symptoms, arrhythmias, or imaging evidence of thrombus, the substantial aneurysmal size in conjunction with this procoagulant genetic profile provided a compelling indication for surgical intervention. This approach aimed to prevent potentially life-threatening complications, including thromboembolic events, aneurysmal rupture, coronary artery compression, and malignant arrhythmias. Although conservative management with anticoagulation has been described in selected cases, the scarcity of long-term follow-up data, the absence of standardized treatment protocols, and the uncertainty regarding the natural history of LAAA in the context of hereditary thrombophilia, particularly in pediatric patients, limit the ability to recommend this approach with confidence. In this context we considered that surgical resection represented a reasonable and potentially definitive option, aiming to reduce the risk of severe complications while acknowledging the need for further evidence to establish optimal management strategies.” (lines 393-418). We are grateful for your constructive feedback, which has allowed us to improve the clarity, relevance, and overall quality of the manuscript.

4. Response to Comments on the Quality of English Language

Response 1: Thank you for your comments

5. Additional clarifications

No additional clarifications

Round 2

Reviewer 1 Report

Comments and Suggestions for Authors

 Accept in present form

Reviewer 2 Report

Comments and Suggestions for Authors

No additional comments